# Strategies to build trust in the conduct of clinical trials: Stakeholders' views in a qualitative study in Ghana

Samuel Tamti Chatio[1,2]*, John Kuumuori Ganle[3], Philip Baba Adongo[3], Patrick Odum Ansah[1], Engelbert A. Nonterah[1,2], Nathan Kumasenu Mensah[4], James Akazili[2], Ulrike Beisel[5]

1 Navrongo Health Research Centre, Navrongo, Ghana, 2 School of Public Health, C.K Tedam University of Technology and Applied Sciences, Navrongo, Ghana, 3 School of Public Health, College of Health Sciences, University of Ghana, Legon, Ghana, 4 Department of Health Information Management, School of Allied Health Sciences, University of Cape Coast, Cape Coast, Ghana, 5 Institute for Anthropology, Faculty of Cultural Studies, Bayreuth University, Germany

*schatio@yahoo.co.uk

## Abstract

While clinical trials have evolved and improved over time producing significant advances in diagnosis, treatment and prevention of diseases, there are equally key challenges such as feasibility of some clinical trials and most importantly the issue of trust in the conduct of clinical trials. Thus, this study provides scientific evidence to address challenges associated with clinical trials conduct as well as a framework describing appropriate trust building strategies to guide the conduct of future clinical trial studies in Ghana and beyond. The study used qualitative research approach where 48 in-depth and Key informant interviews were conducted with participants between June and August, 2019. The interviews were recorded, transcribed and coded into themes using QSR Nvivo 12 software before thematic content analysis. The results revealed low level of trust in the conduct of clinical trials in Ghana. Participants recommended several trust building strategies to improve trust across the clinical trial cycle. Pre-implementation strategies such as effective stakeholder engagement and strengthening clinical trial regulatory bodies were recommended to build community trust. Implementation strategies such as effective monitoring, addressing issues of untrustworthiness and misconceptions regarding drawing and use of blood samples, improved informed consent procedures as well as post-implementation strategy such as timely feedback to clinical trial communities were highly recommended to build trust in clinical trials conduct. Trust is an important factor affecting clinical trials conduct especially in developing countries. The need to invest in national and community level trust-building activities through appropriate stakeholder engagement and effective monitoring systems by clinical trial regulatory bodies are critical strategies to improve trust in clinical trials conduct.

**Data availability statement:** All relevant data are within the paper and its Supporting Information files.

**Funding:** This was part of a bigger study founded by German Research Foundation, grant number BE 5682/4-1 to UB. The funders did not play any role in the study design, data collection and analysis, decision to publish or preparation of the manuscript.

**Competing interests:** The authors have declared that no competing interests exist.

## Introduction

Emerging diseases are a threat to human existence and public health [1]. The causes of these diseases could be through natural disasters, poor sanitation management, weak health systems and in some cases, transmission from infected animal populations. Some of these diseases spread quickly within populations when they occur, resulting in wider outbreaks and many deaths [2–4]. An example of these infectious diseases is the 2014 Ebola Virus Disease (EVD) outbreak in West Africa and other parts of the world [4]. Also, the current Coronavirus disease (COVID-19), as well as the reappearance of old threats such as Yellow Fever and Cholera are all examples of some infectious diseases of public health concern [5,6]. Nonetheless, such challenges provide opportunities for the advancement of preventive and therapeutic medicine for the management of these diseases. Thus, clinical trials are part of the most visible components of biomedical research initiatives aim at generating robust empirical evidence about treatments or preventive interventions to improve health outcomes [7,8].

Clinical trials have evolved and improved over time leading to significant advances in diagnosis, treatment and prevention. The development of infrastructure in Ghana to support the conduct of clinical trials included the establishment of clinical trial regulatory bodies at the national level such as the Ghana Health Service Ethics Review Committee and Ghana Food and Drug Authority (FDA). These bodies are mandated to regulate the conduct of clinical trials in Ghana, review and approve trial protocols prior to implementation. They are supposed to conduct a close monitoring and scrutiny of trials to ensure safety of trial participants. Also, there are three health research centres located in the northern, middle and southern belts mandated to conduct high quality health research including clinical trial to inform health policy and practices. Over the years, these trials have helped to improve clinical trial infrastructure and capacity of staff at the three health research centres. Infrastructure such as laboratories to perform various clinical tests, clinical trial unit equipped with consulting rooms, specimen collection rooms, office and twenty-four-hour internet access availability.

However, there are key challenges in the conduct of clinical trials such as feasibility of some clinical trials, transparency and most importantly the issue of trust in clinical trials conduct [9–11]. According to Lewicki and colleagues, trust is viewed as self-assured positive expectations concerning another person's conduct whereas distrust is having doubts about another person's conduct [12]. Trust therefore plays an important role in a decision-making process. People tend to associate themselves with other persons whom they consider trustworthy and that happens by placing a level of confidence in someone who will not take advantage over you in this case, the conduct of clinical trials [13]. Lack of trust has been reported as a threat to the conduct of clinical trials [14]. For instance, the negative meaning and rumours associated with the conduct of clinical trials regarding their "experimental" nature often adversely affect trust, acceptability and full involvement of community members in biomedical research [15]. Furthermore, personal risks such as fear of possible harm and perceived side effects of clinical trial medicines have been reported to affect participation in clinical trials [9,16]. In Northern Ghana in particular, it has been reported that misconceptions about drawing and use of blood samples during clinical trials affects the conduct of clinical trials aimed at testing new medicines. Some people perceived that blood samples are taken from clinical trial participants and sold or being used for rituals purposes [17,18].

Research is an essential component in epidemic response, as it is the only way to learn how to improve healthcare and to potentially prevent an epidemic from occurring [7]. Trustworthiness and transparency are considered key elements to ensuring successful implementation of public health measures including biomedical research to control public health emergencies [19]. Despite the vital role clinical trials play in ensuring disease prevention and quality healthcare delivery, lack of trust in the design and implementation of clinical trials could

undermine their successful conduct as was the case in the failed EVD vaccine trial in Ghana [20,21]. The need to understand the perceptions of stakeholders on the role of trust and guidelines to improve clinical trials conduct has become urgent. Thus, this study provides scientific evidence to help address challenges associated with the conduct of clinical trials as well as a framework describing appropriate trust building strategies, which could be a vital reference resource to guide future clinical trial studies in Ghana.

## Methods/materials

### Ethical considerations

The study was reviewed and approved by the Ghana Health Service Ethics Review Committee (approval ID: GHS-ERC007/08/18) before the commencements of the study activities. Written informed consent was obtained from all participants who were interviewed. Participants were informed about the purpose, procedures including the benefits and risks of the study and how they were selected as participants. To ensure confidentiality of information, codes were assigned to study participants and used in all study related documents.

### Study design

The study used qualitative research approach where in-depth interviews (IDIs) and Key informant interviews (KIIs) were conducted with participants between June and August, 2019. Qualitative research methodology is where the researcher collects data, makes interpretations of the meaning of the data and presents the data in textual form [22]. Qualitative research approach provides detailed explanations regarding experiences and perceptions of people on the issue under investigation [22]. Thus, this design was suitable because the study aimed at gaining detailed information on appropriate trust building strategies to improve clinical trial conduct. This was a follow-up study to a broader qualitative research on factors affecting trust in clinical trials conduct in Ghana. (https://doi.org/10.1371/journal.pgph.0001178)

### Study site

This was a multi sited-study conducted in the republic of Ghana, located on the west coast of West Africa and shares boundaries with Togo to the east, La Cote d'Ivoire to the west, Burkina Faso to the north and the Gulf of Guinea, to the south. The interviews were conducted in the Kassena-Nankana East Municipality (KNEM) and Kassena-Nankana West District (KNWD) of the Upper East Region of Northern Ghana, Kintampo in the Bono East Region, Hohoe in the Volta Region and Accra in the Greater Accra region of Ghana (see Fig 1)

### Justification of study sites

The reason for selecting KNEM and KNWD was that the two districts fall under the research activities and catchment area of the Navrongo Health Research Centre (NHRC). The NHRC was established in 1989 with the main aim of conducting high quality demographic and health research to inform health policy and practices. Over the years, the centre has conducted many community-based researches, including clinical trial studies in the area with most of them being phase III and phase IV clinical trials. Thus, Community members including those who participated in clinical trials and opinion leaders were selected and interviewed. Also, some of the key informant interviews with clinical trial scientists and regulators were conducted in the KNEM and KNWD.

The failed ebola vaccine trial in Ghana was planned to be conducted in the Hohoe Municipality. Therefore, the Municipality was selected for this study to explore views of community

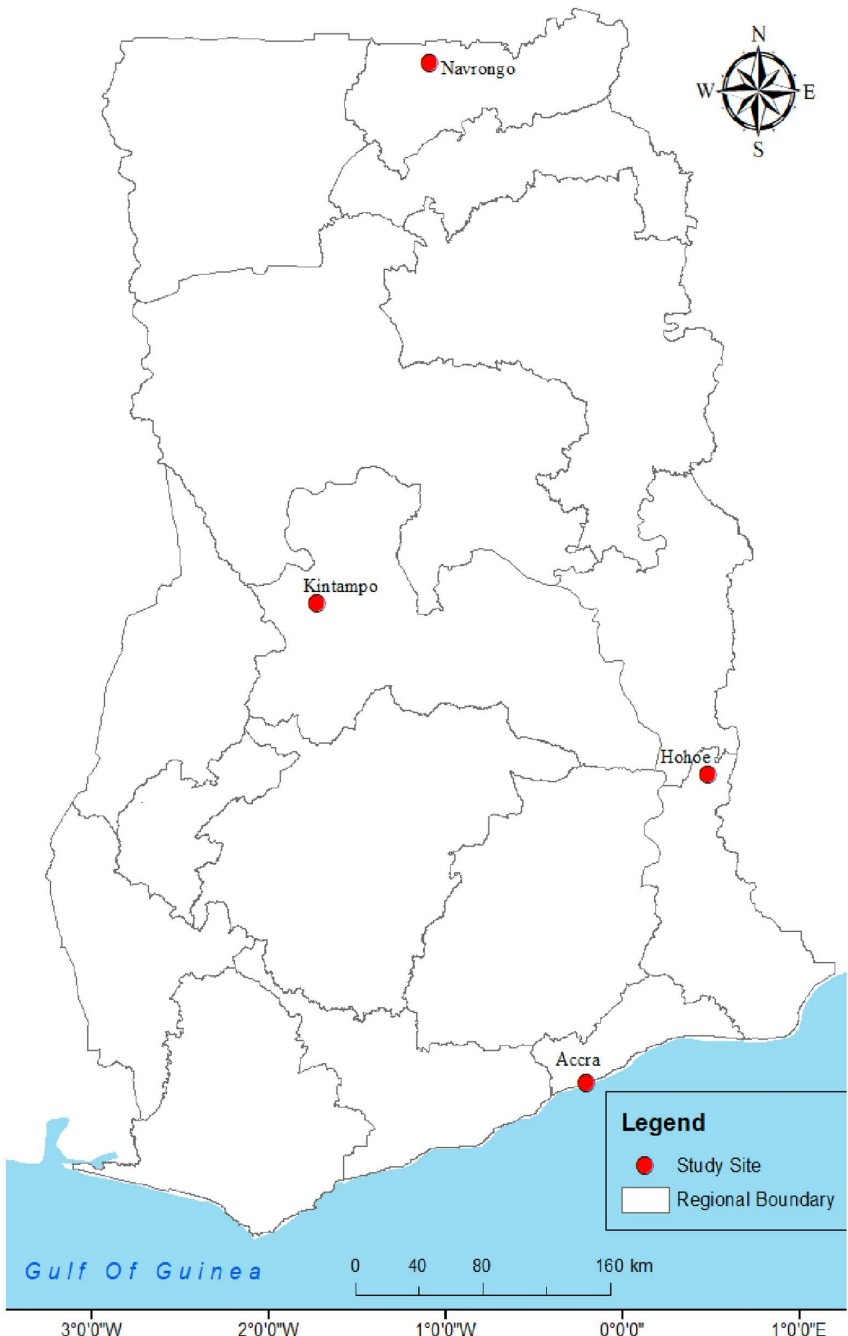

**Fig 1. Map showing the location of the study sites.**

members and opinion leaders who were either consulted and/ or engaged about the failed ebola vaccine trial. Greater Accra Region is the administrative capital of the republic of Ghana where most stakeholders, political leaders including the members of parliament (MPs) live and work. Therefore, the Region was selected as one of the sites where some of the MPs were selected and interviewed. Also, some of the key informant interviews with clinical trial scientists, regulators and monitors were conducted in Accra. Kintampo Health Research Centre

(KHRC) is one of the three health research centres of the Research and Development Division of the Ghana Health Service. The centre was established in 1994 with the main aim of conducting high quality public health and clinical research to influence local and international health policy and programme. Thus, some of the interviews with clinical trial scientists were conducted at KHRC.

## Study population

The study population comprised community members (people who have never taken part in clinical trials, those who qualified but have ever refused to take part in clinical trials, those who were recruited into clinical trials and later dropped out and those who took part in at least 3 clinical trial studies) and opinion leaders such as chiefs, elders and group leaders. Other participants included in the study were clinical trial scientists, monitors and clinical trial regulatory bodies (i.e., ethics committee and Ghana Food & Drug Authority members) as well as members of parliamentary select committee on health.

## Sampling procedures

Non-probability sampling methods such as purposive and snowballing were used in this research. Purposive sampling is where the researcher selects the study participants who can provide appropriate information to help answer the research questions [23]. Snowball sampling is a sampling technique in which existing study subjects provide information to the researcher leading to the identification and recruitment of other participants required for the study [24].

There are different types of purposive sampling methods including maximum variation purposive sampling [25]. This sampling method is used to select participants from different categories with varied experiences and opinions relating to the issue under investigation [25]. Therefore, maximum variation purposive sampling method was used in this study to ensure that a wide range of perspectives relating to the conduct of clinical trials from various groups that matter was captured. However, the snowball sampling method was used to identify some of the community members who qualified but, did refuse to take part in clinical trials and those who were recruited and later dropped out from the clinical trials in the KNEM and KNWD in Navrongo.

## Selection of study participants

**Community members and opinion leaders in Navrongo.** The Navrongo Health Research Centre (NHRC) operates the Navrongo Health and Demographic Surveillance System (NHDSS) in the two districts where fieldworkers routinely visit households within the area to collect and update demographic characteristics of the people including their experiences and involvement in clinical trial studies conducted in the area by NHRC [26]. The list of community members with this experience who participated in at least 3 clinical trials, or qualified but did refuse to take part in clinical trials and those who were recruited into clinical trials and later dropped out was obtained using the NHDSS data. The lead author and a research assistant visited these individuals and those who were willing to take part in the study were included.

Regarding the selection of opinion leaders, two communities (one in KNEM and the other one in KNWD) were selected for the study. The study team visited the selected communities and with the support of chiefs and elders, opinion leaders were identified. The opinion leaders identified were then contacted by the lead author and the research assistant and those who agreed after the purpose of the study was explained to them were interviewed.

### Community members and opinion leaders in Hohoe

In the Hohoe Municipality, purposive sampling technique was also used to select community members and opinion leaders for the interviewers. First, two communities (one community 5km radius within Central Hohoe and the other one 5kms away from Central Hohoe) were selected. The study team then visited these communities and with the support of sub-chiefs and elders in the two communities, list of some community members and opinion leaders was obtained. These people were contacted by the study team and those who agreed after the rationale of the study was explained to them were included in the study.

### Clinical trial scientists, monitors, regulators and members of parliament

Various processes were also used to select these categories of participants, most of which were recruited from the Greater Accra Region. First, official letters were written to the heads of the institutions for their permission to recruit people in their respective institutions into the study. When permission was obtained, the lead investigator subsequently visited these institutions and with the support of the heads, a list of members was obtained. The study team contacted these people using the list obtained and those who were available and willing to participate in the study were included after informed consent was obtained.

### Data collection techniques

Two graduate research assistants were recruited and trained by the first and second authors for data collection. During the training, the research assistants verbally translated the questions contained in the interview guides into the various local languages spoken in KNEM, KNWD (Kasem & Nankani) and Hohoe Municipality (Ewe). This was done to facilitate better understanding and also help data collectors to ask the questions appropriately using these local languages during data collection.

In-depth interview and key informant interview guides were developed by the first and second authors and reviewed by the other co-authors. The guides were finalised by the first author and used to conduct the interviews with participants. The KIIs were conducted with clinical trial scientists, monitors and regulators while the IDIs were conducted with community members, opinion leaders and MPs. Appointments were booked with selected individuals on suitable dates, times and venues before the interviews were conducted. The interviews with the community and opinion leaders were conducted at the community level and venues based on participants' preferences while the interviews with clinical trial scientists, monitors, regulators and MPs were conducted in their individual offices. The interviews lasted between 30–40 minutes, depending on category of participant. English was used to conduct interviews with clinical trial scientists, monitors, regulators and MPs while the main local languages (i.e., Kasem, Nankani and Ewe) spoken in KNWM, KNED and Hohoe were used to conduct the interview with community members and opinion leaders. However, community members and opinion leaders who could fluently speak English opted to be interviewed in English. With consent from participants, all the interviews were audio-recorded using digital voice recorders. A total of 48 interviews (34 IDIs and 14 KIIs) were conducted as detailed in Table 1. The principal of data saturation was used in this study. This means that there was no new information being obtained after the forty-eighth interview.

### Data analysis techniques

The recorded interviews were transcribed verbatim after repeatedly listening to them. Three people with previous experience in qualitative research who could understand the three local languages and English were engaged to transcribe the audio recorded interviews. This was

Table 1.  Summary of the number of interviews conducted in each category.

| No | Category | Data collection method/tool | Number of participants |
|---|---|---|---|
| 1 | Community members who participated in at least 3 clinical trials (Navrongo) | IDI | 6 |
| 2 | Community members who refused/dropped out from clinical trials (Navrongo) | IDI | 6 |
| 3 | Community members who did not take part in trial studies (Navrongo) | IDI | 5 |
| 4 | Community members (Hohoe) | IDI | 5 |
| | opinion leaders (Hohoe) | | 7 |
| 6 | Members of parliament (MPs) | IDI | 5 |
| 7 | Clinical trials scientists & monitors | KII | 7 |
| 8 | Clinical trials regulators (ethics committee and FDA members) | KII | 7 |
| **Total** | | | **48** |

done to avoid bias in the transcription process, which may occur if the same people who conducted the interviews were made to transcribe them. A codebook containing the main themes and sub-themes was developed by the lead author. The codebook was developed deductively using established categories based on the original research questions and objectives, which was validated by the second and fourth authors. The transcripts were then prepared and imported into QSR Nvivo 12 software to facilitate data coding and analysis. The coding process involved a critical review of each transcript and coding of the data into main and sub-themes.

Thematic content analysis was used to analyse the data. The process of thematic content analysis means reading through textual data, identifying themes, coding the themes and then interpreting the content of the themes [27]. The transcripts were prepared and labelled based on variables such as age and category of participant. This method helped the authors to compare views on the issues regarding appropriate trust building techniques presented across the different category of participants and sites during data interpretation. The results were then presented as narrative and supported by relevant quotes from the data.

## Results

### Understanding trust: Participants' perspective

Study participants described trust as being able to understand another person's action and the rationale for such action, in this case the conduct of clinical trials. According to participants, to trust an action or event is to believe that there would be good outcomes while to distrust is to have doubts that an action could produce favourable outcomes.

> To trust means that you need to understand what somebody is doing. So, you need to understand what clinical trials are about because if you do not know the reason why clinical trials are conducted, you will not trust their conduct. **(G-KII-55yr old clinical trial scientist-07)**

> Trust is to believe that someone is genuine and what the person is coming to do will bring about good results. **(IDI-64yr old opinion leader-Hohoe-05)**

Several participants made references to the fact that fake people were many in the system including the scientific field, and this often create distrust including the conduct of clinical trials especially in developing countries.

> You see, you don't know where people are coming from and all of a sudden, they come and say they are doing research. You need to know that they are not "419" people (referring to fraudsters). Some people will come and say they are selling this medicine; they have these

*gadgets to examine you and all kinds of things. There are a lot of fake people in the system and that makes people to have doubts in some of these activities.* **(IDI-59yr old opinion leader-Hohoe-03).**

*My view is that there is so much suspicion in the world. All of us are not very sure about what the next person is doing and that brings about the issue of distrust, which also happens in the conduct of clinical trial studies.* **(IDI-49yr old MP-05)**

Other participants believed that trust was associated with cultural beliefs and norms and depending on beliefs of people, trust or distrust in clinical trials may be high.

*For trust, it will depend on a person's cultural beliefs that may make a person to trust or distrust. So, when you know that this person believes in certain things, if you want to conduct clinical trials and there are taboos in somebody's culture especially when it comes to drawing of blood, it will be difficult to get the person to participate.* **(IDI-59yr old MP-03)**

### Level of trust in clinical trials conduct in Ghana

Participants' views were solicited regarding the level of trust in the conduct of clinical trials in Ghana. The majority of participants, including clinical trial scientists said the level of trust in clinical trials was generally low in Ghana. They gave examples of the failed ebola vaccine trial and the issues people had with the implementation of the piloted malaria vaccine in Ghana to buttress the fact that trust in the conduct of clinical trials was low.

*Trust is low in Ghana. I am saying this because of what happened during the ebola vaccine trial where people did not trust what the researchers wanted to do.* **(IDI-54yr old opinion leader-Hohoe-06)**

*I will say trust is low and if I may give two examples to support my point. You know, the Ebola trial some few years ago that caused the stir and fear in this country. People did not have trust in the conduct of that trial. Secondly, the malaria vaccine trial is another example where you find a number of people still against it. So, these two examples show that our level of trust in these kinds of studies is low.* **(IDI-49yr old MP-04)**

Many participants reported that inadequate stakeholder engagement and community sensitization accounted for the low level of trust. Indeed, some of the clinical trial scientists blamed their colleagues for their inability to adequately engage stakeholders and the larger community on the conduct of clinical trial studies, which had contributed to the low trust.

*I think trust is low and I want to put the blame on some of my colleague scientists because we have not presented our case very well in terms of community and stakeholder engagement.* **(KII-50yr old clinical trial scientist-04)**

*The problem is that, majority of Ghanaians do not know what clinical trial is all about and that is the underlining reason why there is low trust. When it comes to clinical trial research, people think it is just laboratory and when you are now involving human beings, then they say aah why? You want to use us as guinea pigs?* **(KII-38yr old clinical trials monitor-06)**

Some participants particularly, people who were recruited into clinical trials and latter opted out perceived that the level of trust was low because of lack of credibility on the part of some clinical trial scientists.

*When the researchers recruited us into the study, what they (referring to researchers) said they would do, they did not actually do that and that made me to stop being part of the study.*

*The researchers said if the child is sick, the study will take care of the child and they were not doing that.* **(IDI-28yr old community member-dropped out-Navrongo-04)**

*…The researchers said they would take care of you (referring to trial participants), you will not pay anything. The researchers have asked us to pay for all health services and the money would be refunded to us and they were not refunding the money. So, that made me not to trust them.* **(IDI-28yr old community member-dropped out-Navrongo-01)**

## Trust building strategies to improve the conduct of clinical trials

Study participants identified various practical guidelines and strategies that could improve trust in the conduct of clinical trial studies in Ghana. The has been categorized into three main areas: pre-implementation, implementation and post implementation and discussed below.

## Pre-implementation trust building strategies

**Intensive stakeholder engagement.** Intensive stakeholder and public engagement to improve trust in the conduct of clinical trial studies in Ghana was highly discussed by study participants. They held that intensive continuous stakeholder and public engagement and sensitization could enhance knowledge and understanding on the rationale for the conduct of clinical trials. They were of the view that scientists should not wait until there was an opportunity to conduct clinical trials before engaging the public. This strategy was recommended by all the stakeholders in the study.

*I think that there should be continuous public education on clinical trials. Scientists should not wait until there is a clinical trial before educating people about it. People must know that clinical trials are very important and such studies could be conducted in the country as and when necessary.* **(IDI- 54yr old opinion leader-Hohoe-06)**

*If there is any recommendation to be made as we move forward, there should be vigorous public education. We should not wait till there is a clinical trial to be done. It should be part of our public health education process to educate people about clinical trials.* **(IDI-49yr old MP-05)**

Some participants were of the view that people with communication skills should be engaged to educate the general public about the need for clinical trials to be conducted. They added that periodic discussions on issues relating to clinical trials especially benefits and risks on media stations such as radio and television and also during community gatherings could improve trust and participation in clinical trial studies in Ghana.

*…There is the need for discussions at media stations on clinical and why there is the need for clinical trials to be conducted. Also, talking about trust, people need to understand what goes into a trial, who is involved and what will be the outcome of the trial. So, for me, these are the key issues we need to consider to improve trust.* **(KII-55yr old clinical trial scietist-07)**

*I would suggest that before any intervention that would take place at the community level, there should be well-trained people to educate community members very well on radio stations before such interventions are introduced.* **(IDI-32yr old community member-refused to take part in trial-Navrongo-03)**

A good number of participants also suggested that clinical trial scientists should have initial consultation with stakeholders such as politicians, civil society organizations, the media, religious leaders, educational institutions as well as chiefs and opinion leaders anytime there was an opportunity for clinical trial to be conducted in the country. As one of the opinion leaders expressed it:

> *Researchers need to know that the leaders and opinion leaders in every community have about 80% influence over other community members. Therefore, anything that somebody is going to do at the community level including research, if the leaders of that community are informed, I think it will improve trust.* **(IDI-53yr opinion leader-Navrongo-03)**

Other participants especially MPs suggested that the education should focus on evidence whereby researchers use examples from clinical trials conducted in the past to come out with medicines for the management of diseases. That strategy could also improve trust of people and also dispel misconceptions people may have about clinical trials.

> *There should be evidence-based education. Science has advanced and we know that the conduct of trial studies has helped to manage certain diseases in the past. So, if we go and start from there and get a number of vaccines that we know are currently being used, and let people know how these vaccines were developed, it will improve trust.* **(IDI-49yr old MP-05)**

## Thorough protocol review procedures

Views expressed by some participants suggested that thorough review of clinical trial protocols was needed to ensure that the designs of such studies were appropriate and in line with the principles guiding the conduct of clinical trials. They maintained that the review should also focus on clinical trials investigators' qualification, experience and competency.

> *The ethics members should make sure that whoever is doing a clinical trial has the necessary qualification and skills to implement the study. The protocol should be of very good standard.* **(KII-58yr old clinical trial scientist-03)**

> *…If clinical trial regulators do the right thing in the review processes and not just rush to approve protocols especially clinical trial protocols, it would ensure safety and trust.* **(IDI-47yr old MP-01)**

Some community members and opinion leaders reported that clinical trial regulators were the reference point for community members in terms of their role in reviewing and approving these studies, therefore, safety of study subjects should be their ultimate aim to ensure trust of community members in such studies.

> *Actually, the ethics people have very important role to play because as I am seated here, before I will take part in a research study, I have to first of all find out whether these people (referring to ethics committee members) are aware of the research. I probably may want to find out from Food and Drugs Authority to know whether they have heard that this research is going to be conducted or not? So, ethics members have very important role to play in terms of giving permission for researchers to conduct such studies.* **(IDI-60yr Old opinion leader-Hohoe-01)**

## Strengthening clinical trial regulatory bodies

A few participants also called for clinical trial regulatory bodies to be strengthened to enable them play their role effectively, which could contribute to building trust and confidence in the conduct of clinical trial studies in Ghana. They suggested that building the capacity of clinical trials regulatory bodies could help them in the review processes and also ensure that ethical guidelines were strictly followed by clinical trial scientists. They recommended regular training workshops for regulatory bodies in all aspects of clinical trial designs to both improve their skills and keep them abreast with current trends and standards regarding the conduct of clinical trials. These views were mainly expressed by clinical trial scientists as demonstrated in the following excerpts:

> *I will recommend that clinical trial regulators should undergo continuous training to build their capacity because the environment keeps changing. Both FDA and the ethics committee members should be trained so that they can be on top of issues.* **(KII-58yr old clinical trial scientist-03)**

> *The thing is that we have not built the capacity of the ethics committee members and that is a problem. For me, we need to build their capacity to be able to make sure that things are done right. We should intensify the regulations and make the ethics boards stronger to enable them play their supervisory role effectively. This could improve stakeholders' trust in clinical research.* **(KII-50yr old clinical trial scientist-04)**

## Implementation trust building strategies

**Trustworthiness.** Study participants called on clinical trial scientists to be trustworthy in their dealing with community members especially individuals recruited into clinical trial studies. They held that it was important for clinical scientists to be trustworthy by providing the necessary information about clinical trials including benefits and risks to participants and also fulfilling all promises made to participants prior to their enrolment into such studies. They added that it was not about making a lot of promises to convince people to participate, but how such promises were fulfilled could improve trust of community members. These views were expressed mainly by individuals who dropped out from past clinical trials because they felt clinical trials scientists were not honest.

> *For me, if you are doing something with people, just be sincere so that it does not look like you are just using something to convince them. If you (referring to researchers) can do one thing, just say that this is what you can do and when you do it, it will make people trust you.* **(IDI-28yr old community member-dropped out-Navrongo-01)**

> *It is the honesty I will talk about and what I will say is that researchers should try and fulfil their promises. This will motivate people to partake in research.* **(IDI-28yr old community member-dropped out-Navrongo-04)**

Clinical trial scientists, regulators and monitors also urged clinical trial scientists to be trustworthy in all their dealings with community members especially people recruited into clinical trials in terms of the kinds of promises made and how to fulfil them. They advised clinical trial scientists to be transparent, sincere and also stick to research protocols as well as ethics guidelines. Participants believed that this could be the best approach to building trust and confidence of people in the conduct of clinical trials.

> *It is about time that scientists are truthful and build that credibility such that whatever they tell study participants that they will do, they should try and do exactly that. I*

*think when that is done, it will go a long way to build trust.* **(KII-48yr old clinical trial scientist-02)**

*Research scientists should conduct clinical trial studies with sincerity and transparency because those who are taking part in these trials could be their relatives or children and what they (referring to researchers) will not expect to be done to their children, they should not do it to other people who have consented to take part in these trials.* **(KII-38yr old clinical trials monitor-06)**

## Improved informed consent procedures

Study participants suggested that strengthening informed consent procedures through adequate training could improve knowledge and skill of clinical trials field staff to adequately engage and administer informed consent to participants, which could contribute to enhancing their trust and participation in clinical research.

*Researchers (referring to data collectors) should take their time and explain things very well for us to understand and even if a question is asked and they (referring to trial field worker) do not have the answer, they can call the office and ask to enable them explain to the person.* **(IDI-38yr old community member-participated in at least 3 trials-Navrongo-01)**

*….Consenting process has to be done very well and not just telling people about the benefits of a study but also let them know that they are contributing their time to help find medicines to help humanity. So, consenting process should be done very well in order to build trust of community members.* **(KII-48yr old clinical trial monitor-05)**

## Effective monitoring strategies

The need for clinical trial regulatory bodies to strengthen their monitoring strategies by making frequent follow-up visits to trials sites to supervise the activities of investigators was highly recommended by many of the participants. They said that clinical trial regulators should not just approve research protocols, they should have effective monitoring plans to ensure that the rules and regulations guiding the conduct of clinical trials were adhered as demonstrated by views shred by some of the participants in the following excerpts. views

*The ethics people have a role to protect the rights of people who take part in these trials. They (referring to ethics committee members) should not think that because they have given clearance means whatever researchers are doing will not harm participants. If they (ethics committee members) follow-up and make sure that the right thing is done, then they are helping to build community trust.* **(IDI-49yr old MP-04)**

*When the trial starts, FDA should do monitoring to make sure that investigators go strictly according to the research protocol.* **(IDI-49yr old-opinion leader-Hohoe-07)**

Clinical trial scientists particularly described clinical trial regulators as referees of the game, and that regulators had a duty to make sure that study participants' concerns were addressed and their rights protected.

*Ethics committees must also move up a bit in their game to get to the investigators to find out what actually is happening on the field. Also, ethics members must go to the communities and find out if people have issues because they (referring to ethics members) are like the middle men and the referee of the game.* **(KII-53yr old clinical trial scientist-01)**

### Post implementation trust building strategy

**Feedback.** The main post implementation strategy that could build trust of community members in clinical trial conduct was feedback of clinical trial findings according to participants. These individuals held that providing feedback or disseminating findings at the end of clinical trial studies to stakeholders and community members could improve trust in clinical trial conduct in such communities, in this case, Ghana. As one of the community members put it:

> *It is the feedback that is the problem. I think that when researchers are able to come back to us and tell us what they found in the study, people will believe them and they will always be willing to support their work.* **(IDI-30yr old community member-participated in 3 trials-Navrongo-04)**

## Discussion

This study explored views of stakeholders on the issue of trust in the conduct of clinical trials and their ideas on strategies for building trust in clinical trials conduct in Ghana. The findings revealed varied sentiments expressed on the level of trust in the conduct of clinical trials in Ghana, with most participants reporting low trust. These views were supported largely by the indefinite suspension of the ebola vaccine trial and the failure of some parents to allow their children to receive the piloted malaria vaccine in Ghana. The results suggest that inadequate stakeholder engagement largely accounted for the low level of trust in clinical trials conduct in Ghana.

The discussion in relation to the role of mistrust in clinical trial conduct could undermine a successful implementation of biomedical research especially in low and middle-income countries such as Ghana. Thus, this suggests a need to focus attention on confidence building strategies to improve clinical trials conduct in these settings. This paper highlights a number of strategies to address mistrust, and to help enhance the likely success of clinical trial conduct in resource-poor settings, especially in Ghana. Fig 2 summaries key strategies that participants in the current study suggested could be used to build trust in the conduct of clinical trial, which has been categorised into three main areas; Pre-implementation, Implementation and Post-implementation.

At the pre-implementation stage, the focus will be on issues regarding intensive stakeholder engagement to create awareness and improved clinical trial infrastructure could be an effective trust building strategy in the conduct of clinical trials in Ghana (see Fig 2). At the implementation stage, addressing issues of trustworthiness, local cultural concerns and misconceptions regarding drawing and use of blood samples as well as capacity building training of clinical trial staff to improve informed consent administration could help build community and stakeholders' trust. Furthermore, strengthened clinical trial regulatory bodies through training, effective review of protocols and monitoring to ensure compliance and safety of participants, were recommended strategies for building trust in the conduct of clinical trials. At the post-implementation level, dissemination and feedback to community has also been recommended (see Fig 2).

Study participants identified effective and continuous community and or stakeholder engagement by researchers especially biomedical scientists as an essential element to build trust in the conduct of clinical trials. Participants believed that if stakeholders such as community members, patients/potential trial participants, policy makers and civil society organizations are engaged, it could enhance their knowledge and understanding about the conduct of clinical trials. More importantly, periodic discussions on issues concerning the conduct of clinical trials focusing on procedures, benefits and risks on various media stations and during

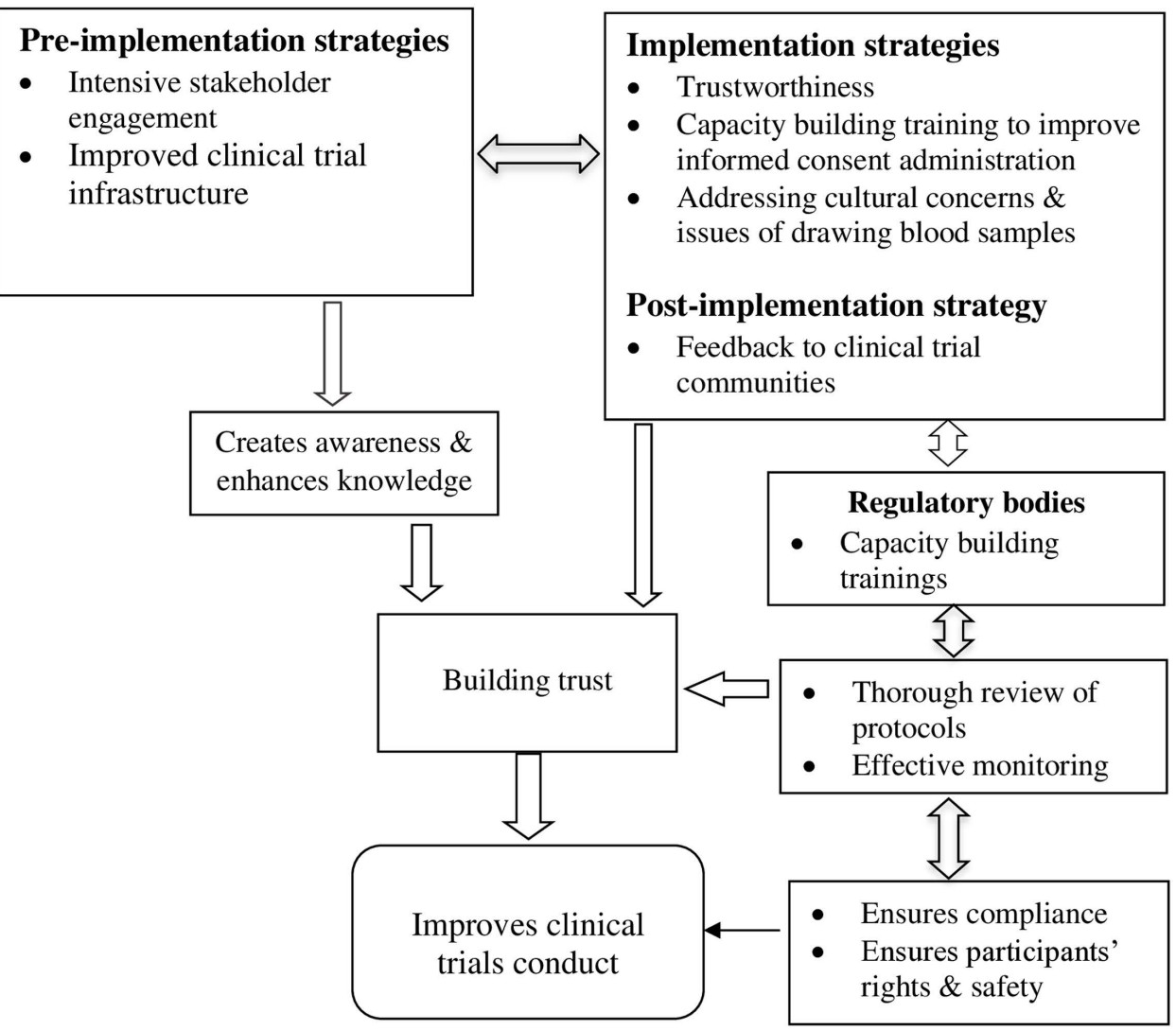

**Fig 2. Proposed trust building strategies to improve the conduct of clinical trials.**

meetings at the community level have been highly recommended to improve trust in the conduct of clinical trials. Additionally, providing evidence-based education on clinical trials to stakeholders both at the community and national levels by clinical trial scientists and regulatory bodies could help dispel the misconceptions people may have about clinical trials, especially the issue of being used as guinea pigs. Participants however did not mention the source of funding for the community and stakeholder engagement activities to create awareness. However, it is important for researchers especially biomedical scientists to take this suggestion seriously to help build trust in the conduct of clinical trials in Ghana. Further, biomedical scientists could add the cost of community engagement to clinical trial budget and justify the need for these activities to be carried out, which is merely to create community awareness, build their trust and confidence in clinical trial conduct.

Community engagement involves interactive relationship between researchers, policy-makers and the community where people are seen as partners in research rather than merely subjects. Focusing on the need to conduct biomedical studies through community education could provide

valuable input in identifying ways to improve trust in the conduct of clinical trial studies [11,28]. Quite apart from that, community engagement will build capacity of people to understand the research process, raise concerns and ultimately help find solutions to unexpected issues that may arise in the process of the research [29]. Where researchers fail to genuinely and appropriately engage communities at the pre-implementation stages of the research, mistrust may be engendered in the process, and this may affect timely conduct and completion of research studies [30]. This suggests that effective engagement with key stakeholders such as political leaders, the media, civil society organizations, religious leaders as well as chiefs, elders and opinion leaders at the community level at the initial stages could have facilitated the conduct of past clinical trial studies that were not successful in Ghana including the ebola vaccine trial study.

Notwithstanding the benefits of community engagement in community-based projects including biomedical studies, there are also a number of challenges that hinder effective community engagement exercise. For instance, history of community exploitation by researchers, lack of interest by community members and cultural differences between community and researchers have been highlighted as key challenges associated with community engagement processes [31]. Therefore, researchers need to take pragmatic steps to deal with these challenges. More importantly, researchers need to understand that community engagement must meet the needs of the populations and/or communities affected by the research, strengthening the community's role and capacity to actively address research priorities and helping to ensure the development and implementation of relevant, feasible and ethical research. This will help build mutual understanding and trust thereby helping to improve relationships between researchers and community members to improve the conduct of clinical trial studies [32].

At the implementation stage, the role of clinical trial regulators in the conduct of clinical trials has been viewed as very important to ensuring that the rights and safety of community members especially direct study participants are protected. Findings of this study particularly point to the fact that clinical trial regulators are the referees in terms of their role in reviewing and approving clinical trial protocols and also the recruitment of community members into these studies. Therefore, the rights and safety of study participants should be their ultimate aim to help improve trust in the conduct of clinical trials in Ghana. Thus, thorough review of clinical trial protocols and effective monitoring of the activities of clinical trial scientists especially at the implementation stage, trust and confidence could be built. This will also help to address the issue of dishonesty and unreliability on the part of clinical trial scientists and also reduce outrageous and hostile behaviours of some clinical trial staff (see Fig 2). Thorough review of clinical trial protocols with particular attention to qualification and competency of clinical trial staff before approval is highly recommended to help address the issue of trust in clinical trials conduct. This recommendation is consistent with previous studies, which noted that strict regulations and clear system of accountability make people feel safe in biomedical studies [33]. However, effective regulation cannot happen without strengthening clinical trial regulatory bodies in this case, ethics review committees and food and drug Authority mandated to regulate biomedical research activities in Ghana. Thus, an important step is to improve the knowledge and skills of regulatory institutions to appropriately regulate and monitor the conduct of research studies in Ghana especially clinical trials. Evidence exists that lack of experienced regulatory staff is a challenge to the conduct of clinical trials [34,35]. This is why participants in this study noted that periodic training in all aspects of clinical trial designs is required to keep regulatory bodies abreast with current trends and standards regarding the conduct of clinical trials. The Ghana Health Service Ethics Review Committee and Ghana Food and Drug Authority are national institutions mandated to regulate the conduct of clinical research. Therefore, the Government of Ghana could allocate money from the national budget to take care of the training of these regulatory bodies in Ghana.

The issue of trustworthiness has been reported in earlier study conducted in Ghana as an important factor influencing trust of people in the conduct of clinical trials at the implementation stage [36]. Where clinical trial scientists are being honest by providing the right and appropriate information in all aspect of clinical trials, it could improve trust and participation. As part of the ethical guidelines for conducting clinical trials, the necessary information including benefits and risks about research are supposed to be known to potential participants to enable them take a decision whether to take part or not. Failure on the part of clinical trial scientists to do this could lead to mistrust on the part of community members if they realize that scientists are not being truthful. This means that professional integrity and reputation is paramount in building community trust and confidence in research studies especially clinical trials conduct. Thus, it is important for research scientists to be trustworthy and transparent in their dealings with community members especially potential participants to help build their trust in the conduct of clinical trials [33,37].

Lack of knowledge of local culture and customs by clinical trial scientists can negatively affect the conduct of clinical trial studies [38]. It is essential for researchers, particularly biomedical researchers to understand the social and cultural context of the community in terms of its trends and history regarding research [39]. Related to the issues of cultural values and norms are the apprehensions regarding drawing and use of biomedical samples such as blood that affects trust in clinical trials conduct [36]. These views are apparently fuelled by weak community engagement and poor consenting procedures especially on drawing and use of blood samples from participants in clinical trials. Informed consent administration is viewed as very vital in the conduct of clinical trials studies and depending on how well clinical trial staff are able to engaged and explain study procedures to potential participants could influence their trust [40]. Therefore, improved informed consent administration through continuous training to build the capacity of clinical trial staff and improved community-based disseminations or feedback as post-implementation strategies could be key to ensuring dispelling wrong notions as well as building and sustaining community trust, interest and support in the conduct of clinical trials (see Fig 2). These proposals support findings from previous studies recognizing that training at different levels for clinical trial investigators and other staff on good clinical trials practices could enhance community confidence and participation in clinical trial studies [41,42].

## Study limitations

The study has the following limitations. One limitation of this study is that some of the interviews with community members were conducted in the local languages (Kasem, Nankani, and Ewe), tape recorded, transcribed and translated into English. It is possible that the meaning of some statements made in the local language may have been lost in the English translation. However, the interviews were transcribed by people who were native speakers with experience in transcribing qualitative interviews. Any loss of meaning during the translation was thus minimized, thus did not affect the findings of the study. Another limitation is that since the study used purposive sampling, a non-probability sampling method and also a small sample size, the views expressed by study participants are their personal opinions and may not necessarily represent the views of the larger population.

## Conclusion

Trust is an important factor affecting clinical trials conduct especially in developing countries such as Ghana. However, building public trust and confidence in the design and implementation of clinical trials especially in developing countries where the burden of new and emerging diseases especially infectious diseases is often high remains a daunting challenge.

Thus, there is need for collaborative efforts by all stakeholders including research institutions, clinical trials regulatory bodies, civil society organizations as well as research grant awarding organizations to take the issue of trust in the conduct of clinical trial studies seriously. These findings suggest the need for investment in national and community level trust-building activities through appropriate community and or stakeholder engagement strategies. As recommended in this study, effective monitoring systems by clinical trial regulatory bodies, improved informed consent procedures, trustworthiness on the part of clinical trial scientists and institutions are critical strategies to improve trust in the conduct of clinical trial studies especially in low and-middle-income countries such as Ghana.

## Supporting information

**S1 Data.  XXXX.**
(DOCX)

**S2 Data.  XXXX.**
(DOCX)

**S1 Checklist.  XXXX.**
(DOCX)

## Acknowledgement

The authors wish to thank all the study participants who shared their views and experiences on the role of trust in clinical trials and trust building strategies to improve the conduct of biomedical studies in Ghana. The authors also wish to acknowledge the contribution of research assistance (Mr. Isaac A. Ayaga and Miss. Enyonam Duah) who helped us during data collection.

## Author contributions

**Conceptualization:** Samuel Tamti Chatio, John Kuumuori Ganle, Philip Baba Adongo, Ulrike Beisel.

**Data curation:** Samuel Tamti Chatio.

**Formal analysis:** Samuel Tamti Chatio, Engelbert A. Nonterah, Nathan Kumasenu Mensah, James Akazili.

**Funding acquisition:** John Kuumuori Ganle, Ulrike Beisel.

**Investigation:** John Kuumuori Ganle, Ulrike Beisel.

**Methodology:** Samuel Tamti Chatio, John Kuumuori Ganle, Philip Baba Adongo, Patrick Odum Ansah, Engelbert A. Nonterah, Nathan Kumasenu Mensah, James Akazili, Ulrike Beisel.

**Project administration:** John Kuumuori Ganle, Ulrike Beisel.

**Resources:** John Kuumuori Ganle, Patrick Odum Ansah, Ulrike Beisel.

**Software:** Samuel Tamti Chatio, Engelbert A. Nonterah, Nathan Kumasenu Mensah.

**Supervision:** John Kuumuori Ganle, Philip Baba Adongo, Patrick Odum Ansah, James Akazili, Ulrike Beisel.

**Validation:** Engelbert A. Nonterah, Nathan Kumasenu Mensah, James Akazili.

**Visualization:** Samuel Tamti Chatio, Philip Baba Adongo, Patrick Odum Ansah.

**Writing – original draft:** Samuel Tamti Chatio.

**Writing – review & editing:** John Kuumuori Ganle, Philip Baba Adongo, Patrick Odum Ansah, Engelbert A. Nonterah, Nathan Kumasenu Mensah, James Akazili, Ulrike Beisel.

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
