## [Decision Letter · Decision Letter 0]

19 Jun 2024

PGPH-D-24-00458

Trust building strategies to improve the conduct of clinical trials. stakeholders’ views in a qualitative study in Ghana

Dear Dr. Chatio,

Thank you for submitting your manuscript to PLOS Global Public Health. After careful consideration, we feel that it has merit but does not fully meet PLOS Global Public Health’s publication criteria as it currently stands. Therefore, we invite you to submit a revised version of the manuscript that addresses the points raised during the review process.

Please carefully revise your manuscript following the suggestions from Reviewer 2, particularly the need to include further details about the interviews and clarify the results.

We look forward to receiving your revised manuscript.

Kind regards,

Jennifer Tucker, PhD

Staff Editor

Journal Requirements:

that supported your study, including funding received from your institution. 

3. In the online submission form, you indicated that "The data used for this manuscript are available, and will be shared with the editor

based on request.". 

3. Uploaded as supplementary information.

4. Please provide separate figure files in .tif or .eps format.

5. Tables should not be uploaded as individual files. Please remove these files and include the Tables in your manuscript file as editable, cell-based objects. For more information about how to format tables, see our guidelines:

https://journals.plos.org/globalpublichealth/s/tables

Additional Editor Comments (if provided):

Reviewers' comments:

Reviewer's Responses to Questions

**Comments to the Author**

1. Does this manuscript meet PLOS Global Public Health’s publication criteria ? Is the manuscript technically sound, and do the data support the conclusions? The manuscript must describe methodologically and ethically rigorous research with conclusions that are appropriately drawn based on the data presented.

Reviewer #1: Yes

Reviewer #2: Partly

2. Has the statistical analysis been performed appropriately and rigorously?

Reviewer #1: Yes

Reviewer #2: N/A

3. Have the authors made all data underlying the findings in their manuscript fully available (please refer to the Data Availability Statement at the start of the manuscript PDF file)?

Reviewer #1: Yes

Reviewer #2: Yes

4. Is the manuscript presented in an intelligible fashion and written in standard English?

Reviewer #1: Yes

Reviewer #2: Yes

5. Review Comments to the Author

Reviewer #1: well written manuscript

logical and need for trust of clinical trial process in present situations

Implementation of these policies dire required for gaining trust of people involved

Most of the clinical trial scenario is wisely presented in the article

I would appreciate the presentation of facts in the article

Reviewer #2: The Abstract is clearly presented and accurately outlines the study to be presented.

Introduction: is brief but sets the scene, relating to clinical trials in Ghana, for the study to follow. Having read the entire paper I would suggest that this needs to include more background information about the infra-structure to support the conduct of trials in Ghana as what is being proposed about the challenges of conducting research in Ghana are actually globally applicable, it’s the solutions that appear to need to be introduced/ refined.

Methods: these are clearly and thoroughly explained with the exception of the Interview Schedule. We are only told that ‘interview guides were developed and used to conduct the interviews…’ How were they developed, who developed them, was there stakeholder involvement in this process?

The schedule needs to be included, possibly as supplementary material, (unless I missed it) as it is not possible to gauge whether the interview questions were neutral or biased. The authors clearly wish to make the point about trust being important in decision making but there are other factors that might also impact on decision making yet we do not know if they were included. After all the diligence given to the conduct of the study this is an important omission.

Data analysis: appears to be thorough and guided with reference to established methods.

Results:

Many of the points raised in the results are not specific to Ghana. It is well known that recruitment is a difficult aspect of running a clinical trial and that many studies are eventually seriously under-powered or indeed stopped due to failure to recruit. While the context is different the issues are the same. Why would any potential participant take part in a study if they did not trust the researchers?

Page 13 reports the need for intensive stakeholder engagement and this applies to all studies that involve medicinal products. There is no mention of PPI – patient and public involvement, the importance of which has been emphasised in research for many years.

(Patient and public involvement policy | Public Involvement Programme | Public involvement - putting you at the heart of our work | NICE and the public | NICE Communities | About | NICE).

The programme of research relating to interventions such as the Qunitet Recruitment Intervention (QRI) have been developed in recognition that many studies require specific inputs and processes in order to recruit to target.

(Optimising recruitment and informed consent in randomised controlled trials: the development and implementation of the Quintet Recruitment Intervention (QRI) - PubMed (nih.gov)).

These are just two examples of the wider recognition that stakeholder involvement is widely known to be vital in clinical research.

I think that the more important issues are that in Ghana, what are the solutions and what are the barriers and the facilitators?

P14- recommends the need for public education for clinical trials- I think the important issue here is whose responsibility is it and who will pay for it rather than that it is required.

Similarly ‘…people with communication skills should be engaged to educate the general public about the need for trials to be conducted’ the more interesting question is who should be doing this in Ghana and how and who will fund it?

P15- lines 366- 370 - this quote indicates the need for Steering Groups involving opinion leaders, this would be expected in a trial, so the pertinent issue is whether they (SGs) are part of the study protocol in Ghana and if not then they need to include them.

P16 deals with ‘Thorough protocol review procedures’ this puzzles me as surely in order to get funded these clinical trials are reviewed thoroughly. How can the researchers get their study funded without rigorous review?

The need for ethical approval is raised and again this is a stage that is mandatory for clinical research – I am wondering how have these researchers been able to proceed without ethical approval? There other issues in this section relate to research governance and regulatory processes and these are widely regarded as standard requirements, if they are not part of the research scene in Ghana then I think that a more interesting paper is what is to be done to get them established.

P17 – Line 416 – ‘I will recommend that clinical trial regulators should undergo continuous training …’ once again I don’t think anyone would disagree with this statement, the issue is why is it not happening in Ghana? What are the barriers and facilitators to such training? Good clinical practice certificates are required of researchers in the Uk to be renewed every three years and I believe there are similar requirements in other countries, it would seem that these are also needed in Ghana. If they are not required, what needs to happen for them to be part of the research culture.

P 18 – lines 432 – 434 – ‘ …it was not about making a lot of promises to convince people to participate ….’ This would be expected and clearly explained in patient information documents / tools etc.

Line 449 – ‘They advised research scientists to be transparent, sincere and also stick to research protocols as well as ethics guidelines’ This is the expectation, the more relevant question is if it is not happening then why not and how to change the culture.

P 19 Effective monitoring strategies are called for, as above researchers understand the need for these, that is why there are independent trial monitoring committees, why there is the need for fidelity to the protocol and trial process monitoring.

Similarly in other aspects of the results- the question is not whether these are important, it is why are they not in place in Ghana, what are the barriers and how can they be overcome?

Discussion – to make this paper have more novel it needs to deal more thoroughly with how can all these valuable points raised in the interviews be implemented in ‘…resource poor settings, especially in sub-Sahara Africa’. The issues that are raised from the interviews such as the need for trust in the research/ers, the need for ethical approval, for trial monitoring, for dissemination to communities etc apply more widely than just in Ghana. For example, the MRC Guidance (A new framework for developing and evaluating complex interventions: update of Medical Research Council guidance - PubMed (nih.gov)) includes ways of planning to deal with many of the important tissues raised in this study – this indicates that these issues are widely applicable. However, the means to overcome them in a country with little resource is particularly challenging. It could be argued that the research should not be funded until the research infra-structure is in place to promote all these important issues raised by the authors.

6. PLOS authors have the option to publish the peer review history of their article (what does this mean? ). If published, this will include your full peer review and any attached files.

**Do you want your identity to be public for this peer review?** For information about this choice, including consent withdrawal, please see our Privacy Policy .

Reviewer #1: No

Reviewer #2: No

---

## [Decision Letter · Decision Letter 1]

28 Oct 2024

PGPH-D-24-00458R1

Trust building strategies to improve the conduct of clinical trials. stakeholders’ views in a qualitative study in Ghana

Dear Dr. Samuel Tamti Chatio,

Thank you for submitting your manuscript to PLOS Global Public Health. After careful consideration, we feel that it has merit but does not fully meet PLOS Global Public Health’s publication criteria as it currently stands. Therefore, we invite you to submit a revised version of the manuscript that addresses the points raised during the review process.

We look forward to receiving your revised manuscript.

Kind regards,

Vishal Goyal

Academic Editor

Journal Requirements:

Reviewers' comments:

Reviewer's Responses to Questions

**Comments to the Author**

Reviewer #1: 

1. The manuscript is technically sound and it provides new knowledge in the area of publication

2. Consider recasting the title to something like.. strategies to build trust.....

3. There seems to be an over emphasis on infectious diseases as the main basis for clinical trials. It would be better to broaden the usefulness of clinical trials to other areas such as non communicable conditions.

4. Line 176, Similar strategy was used in Hohoe Municipality ....is inaccurate because the selection strategy in Navrongo was different from that used in Hohoe

5. May need to declare possible bias in respondents as only those available and willing were recruited. Those unavailable and unwilling to participate could potentially skew the findings. This should be declared as a study limitation

6. It is not clear what verbal translation of the interview guides means. Were the tools systematically translated before data collection? If not, this may need to be declared as a study limitation as the questions could change with different interviewers translating during data collection.

7. Check editorial typos in lines 12, 163. 176, 180, 351, 364

Reviewer #2: 

1. The opening remarks should start with describing what is trust; and a bit of describing what is clinical trials. This is enabling the reader to link the two key issues.

2. In addition, we have noted trust can also be achieved by clinical trials Sponsor organizations clinical monitoring teams during onsite monitoring of clinical trials and site teams trainings/support. They build trust with regulators and site staff in terms of trusting the data collected and processes.

3. Misconceptions on drawing blood in northern Ghana as referenced on 17,18 – please add example for the readers to link those misconceptions and the research results.

4. Study design and site selection - please provide more details why participants were drawn from those particular areas and not randomly across the country, to provide a good representative sample, and avoid bias. May be justification around budget constraints and logistics may suffice. Add a map of Ghana and indicate the study area to the paper. This will make the reader to a clear view of sampling approach. An attempt to conduct phone interview with those leaders and researchers away in rural parts of Ghana may have provided a rich information around remote areas perceptions.

5. Study population – please provide more details on the composition of the opinion leaders reached out . whether they are church leaders, community advisory board members, local administration etc. This will provide readers with an opportunity to determine the study population of interest if techy want to replicate the study,

6. Bullet 204 – letters MPs has been used but not defined anywhere else within the paper. Please provide full names for the acronym.

Generally the study and paper well done. The researchers may in future want to know the if the clinical trials centres staff have trust with studies they are been hired by Investigators to conduct ; and also the Investigators views on the studies they are conducting on behalf of the Sponsor companies. This is because, you can inspire confidence and trust on others when yourself not confident in a process. Most of these site staff and Investigators may be motivated by money ; and may not trust what they involved in.

---

## [Decision Letter · Decision Letter 2]

12 Mar 2025

Strategies to build trust in the conduct of clinical trials. Stakeholders’ views in a qualitative study in Ghana

PGPH-D-24-00458R2

Dear Dr Chatio,

We are pleased to inform you that your manuscript 'Strategies to build trust in the conduct of clinical trials. Stakeholders’ views in a qualitative study in Ghana' has been provisionally accepted for publication in PLOS Global Public Health.

Best regards,

Ferdinand C Mukumbang, PhD

Academic Editor

Reviewer Comments (if any, and for reference):

Reviewer's Responses to Questions

**Comments to the Author**

1. If the authors have adequately addressed your comments raised in a previous round of review and you feel that this manuscript is now acceptable for publication, you may indicate that here to bypass the “Comments to the Author” section, enter your conflict of interest statement in the “Confidential to Editor” section, and submit your "Accept" recommendation.

Reviewer #4: All comments have been addressed

Reviewer #5: All comments have been addressed

2. Does this manuscript meet PLOS Global Public Health’s publication criteria ? Is the manuscript technically sound, and do the data support the conclusions? The manuscript must describe methodologically and ethically rigorous research with conclusions that are appropriately drawn based on the data presented.

Reviewer #4: Yes

Reviewer #5: Yes

3. Has the statistical analysis been performed appropriately and rigorously?

Reviewer #4: Yes

Reviewer #5: Yes

4. Have the authors made all data underlying the findings in their manuscript fully available (please refer to the Data Availability Statement at the start of the manuscript PDF file)?

Reviewer #4: Yes

Reviewer #5: Yes

5. Is the manuscript presented in an intelligible fashion and written in standard English?

Reviewer #4: Yes

Reviewer #5: Yes

6. Review Comments to the Author

Reviewer #4: Have reviewed my comments and all of them have been answered.

Reviewer #5: The elaborate need for improving conduct of clinical trials in Ghana has been presented well. The suggestion to ensure pre, during and post implementation strategies is promising. The practical realities of being able to act on these by engaging with stakeholders and to disseminate the findings to the participants as a method to gain trust needs follow up to observe if feasible in low resource settings. It would be good to see follow up studies if any of the suggestions are implemented. Also, as addressed int he limitations, the geographical setting and purposive sampling needs careful consideration on the validity of the findings from this study.

7. PLOS authors have the option to publish the peer review history of their article (what does this mean? ). If published, this will include your full peer review and any attached files.

**Do you want your identity to be public for this peer review?** For information about this choice, including consent withdrawal, please see our Privacy Policy .

Reviewer #4: **Yes: ** Nick Kisengese

Reviewer #5: **Yes: ** Dr. Somasundari Gopalakrishnan
